# Achieving universal sanitation in Ghana: An analysis of key drivers of toilet ownership among property owners in Urban areas

Godwin Armstrong Duku[1], Nana Kobea Bonso[2]*, Eugene Appiah-Effah[1‡], Yenube Clement Kunkuaboor[3‡], Emmanuel Nouwati[3‡], Barbara Gyapong-Korsah[4‡], Ebenezer Acquah[1‡], Kwabena Biritwum Nyarko[1‡]

**1** Department of Civil Engineering, Regional Water and Environmental Sanitation Centre, Kwame Nkrumah University of Science and Technology (KNUST), Kumasi, Ashanti, Ghana, **2** Department of Civil Engineering, University of Development Studies, Tamale, Ghana, **3** Department of Statistics and Actuarial Science, Kwame Nkrumah University of Science and Technology (KNUST), Kumasi, Ashanti, Ghana, **4** Office of Grant and Research, College of Health Sciences, Kwame Nkrumah University of Science and Technology (KNUST), Kumasi, Ashanti, Ghana

☯ These authors contributed equally to this work.
‡ These authors also contributed equally to this work.
* kwesikobea@yahoo.com, kbonso@uds.edu.gh

**Data Availability Statement:** All relevant data are within the manuscript and its Supporting Information files.

## Abstract

Access to safe sanitation facilities remains a critical public health concern, especially in rapidly urbanizing countries like Ghana. This study investigates the determinants of household toilet ownership among property owners in three urban districts in Ghana. Using a cross-sectional survey design, data were collected from 1,256 property owners selected through a multi-stage stratified sampling procedure. Logistic regression analysis revealed that toilet ownership is significantly associated with the age and education level of property owners, community classification, building characteristics, and household income. Older property owners were more likely to own toilets (OR = 1.014 per year increase), as were those with higher education levels (OR = 1.752 for secondary, OR = 4.489 for tertiary education). Medium-class communities (OR = 2.013) completed buildings (OR = 2.625), and those constructed with sandcrete (OR = 12.755) were more likely to have toilets. Higher household income (OR = 1.00) correlated positively with toilet ownership. We conclude that enforcing building regulations requiring toilet facilities in all properties is crucial for improving sanitation in urban Ghana. Additionally, innovative sanitation financing interventions that subsidize the costs of sanitation facilities can be effective in addressing financial barriers and increasing household toilet ownership.

## Introduction

Sanitation, defined as the safe disposal of human excreta, is a critical determinant of public health. Access to safe and hygienic sanitation facilities, particularly toilets, is crucial in preventing the spread of infectious diseases [1,2]. Poor sanitation practices, such as open defecation,

**Funding:** The author(s) received no specific funding for this work.

**Competing interests:** The authors have declared that no competing interests exist.

contaminate water sources and the environment, significantly increasing the risk of diarrhoeal diseases. These diseases are a leading cause of childhood mortality globally, with approximately 395,000 deaths annually in children under five [3]. Beyond diarrhoeal diseases, inadequate sanitation contributes to environmental degradation and economic challenges [3].

In recognition of the importance of sanitation, the United Nations included access to safe and affordable sanitation in the Sustainable Development Goals (SDGs). SDG 6 aims to "ensure universal access to water and sanitation for all" [4,5]. Despite global efforts, significant disparities remain in toilet ownership and access to improved sanitation facilities, especially in low- and middle-income countries (LMICs). In 2022, an estimated 1.5 billion people still lacked access to basic sanitation services, such as private toilets or latrines [3,6].

Ghana, like many LMICs, is working to improve its sanitation situation. The country has made some progress in sanitation access in recent decades. The National Environmental Sanitation Policy, launched in 2010, emphasizes the importance of toilet ownership and promotes the construction and use of household toilets [7,8]. Additionally, the Ghana Water, Sanitation, and Hygiene Sector Development Programme (GWASHSDP) outlines a comprehensive plan to achieve universal access to sanitation by 2030 [9]. Despite these efforts, only 29% of the population in Ghana has access to basic sanitation services [6]. Urban areas, although generally better off with 34% coverage compared to their rural counterparts with 21% coverage, still face challenges related to inadequate sanitation infrastructure and maintenance issues exacerbated by rapid urbanization.

Ghana's urban population is growing at an annual rate of about 4%, particularly in major cities that attract individuals seeking better economic opportunities [10]. These urban areas are characterized by high population densities, informal settlements, and varying levels of economic development, which collectively strain existing sanitation systems. In urban areas, decent and affordable housing remains a major challenge. Housing shortages have forced some households to use makeshift structures, such as metal containers and kiosks, as sleeping places or to occupy inadequately constructed dwellings. Almost 1 out of every 2 urban households (46%) live in rented houses, yet a significant majority of the urban population lacks access to basic sanitation [11]. This high prevalence of rental housing means that property owners, who often make decisions regarding toilet construction and investment, could be a critical focus for urban sanitation intervention. Furthermore, the high cost of constructing and maintaining toilets, coupled with limited space and tenure insecurity, further hampers efforts to improve sanitation in urban areas [12,13]. In these contexts, the availability and quality of household toilets are not merely matters of convenience but of public health urgency.

Existing studies have primarily focused on open defecation and its determining factors in both rural and urban areas. These studies have identified factors such as poverty, education level, household size, access to improved water sources, mass media, and cultural beliefs as key determinants [14–19]. However, there is a paucity of studies investigating the specific factors influencing toilet ownership among property owners in urban Ghana. This research gap is particularly relevant given the rapid urbanization Ghana is experiencing. The focus on property owners is particularly pertinent as they hold decision-making power regarding toilet construction and investment. Property owners are more likely to invest in permanent and improved sanitation facilities compared to renters, who may have less incentive or capability to make such investments. In Ghanaian urban contexts, where rental housing is prevalent, understanding the decision-making processes of property owners regarding toilet ownership is crucial for developing targeted interventions to improve sanitation coverage.

This study addresses this gap by investigating the key factors that influence household toilet ownership among property owners in three urban districts: the Kumasi Metropolitan Assembly, Ga North Municipal, and Akuapem North Municipal Assembly. These districts fall within

the Ashanti, Greater Accra, and Eastern regions, which are among the most urbanized and economically active areas in the country. Kumasi reflects the dynamics of a densely populated urban center. Ga West, situated in the peri-urban area of the capital, Accra, provides insights into sanitation issues in rapidly urbanizing districts with mixed socio-economic profiles. Akuapem North, though more suburban, offers a contrast to the more densely populated urban centers and helps to capture the diversity of urban experiences across regions. Thus, these districts exhibit varied levels of development, housing structures, and enforcement of sanitation regulations, which allowed for a more comprehensive analysis of the factors influencing toilet ownership in urban Ghana.

## Research methods

### Study settings

The study was conducted in three out of the sixteen regions of Ghana (Fig 1), namely the Ashanti Region, the Eastern Region, and the Greater Accra Region. A district, Akuapem North Municipal, Kumasi Metropolitan, and Ga West Municipal, was selected from the Eastern, Ashanti, and Greater Accra Regions, respectively, through a simple random sampling technique.

Akuapem North Municipality is one of the 32 districts in the Eastern Region of Ghana. It was created as a 'district' in 1988 as part of the Government of Ghana's decentralization policy and then elevated to the status of a 'municipality' in 2012. Akropong, the capital of the municipality, is located approximately 23.7 kilometers south-east of the Eastern Regional capital, Koforidua, and 48.3 kilometers north-north-east of the national capital, Accra. Akuapem North Municipal's population in 2021 was estimated as 105,315. The municipality occupies a land size of 450 km$^2$ with a population density of 234.03 persons per square kilometer [20].

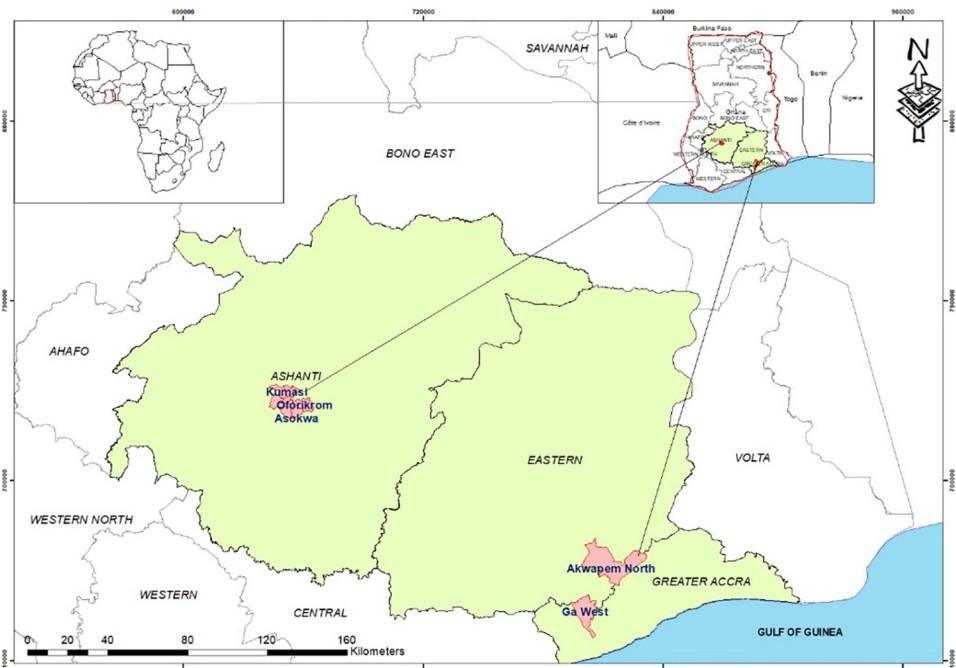

**Fig 1. Map of study areas.** (Source: The map was created using ArcMap 10.8.1 with a shapefile for Africa obtained from Nature Earth and the regional shapefile obtained from OCHA).

Ga West Municipality is one of the 26 administrative districts in Greater Accra Region. The erstwhile Ga District was created in 1988 as part of the Government's decentralization policy. In 2004, Ga District was split into Ga East and Ga West Districts, with Ga West attaining the status of a 'municipality' in 2012. The municipal capital, Amasaman, is located approximately 20.8 kilometers North-North-West of the national capital, Accra. In terms of population, Ga West is one of the fastest-growing municipalities in the region, and its growth is driven by peri-urbanisation. Estimates based on the 2021 Population and Housing Censuses indicates that the population of the Ga West Municipality is 314,299. It occupies a land area of approximately 284.08 square kilometers [21].

The Kumasi Metropolis (or the City of Kumasi) is one of the oldest administrative districts in Ghana. When Ghana's decentralization policy was introduced in 1988, Kumasi, together with Accra and Sekondi-Takoradi, was assigned the status of a 'metropolis' and has since been administered by the Kumasi Metropolitan Assembly. Kumasi Metropolis, which also doubles as the capital of the Ashanti Region, is located almost at the center of the region and is approximately 201.6 kilometers North-West of Accra. The Kumasi Metropolis is divided into four (4) Sub-Metropolitan District Councils namely Bantama, Manhyia, Nhyiaeso and Subin. The population of the Metropolis, according to the 2021 Population and Housing Census, stands at 443,981 people [22].

## Study design and sampling procedure

The study used a cross-sectional quantitative research design to gather data from 1,256 randomly selected property owners. The survey covered 409 property owners in Akuapem North, 423 in Ga West, and 424 in Kumasi Metropolis. The sample size for each of the three study areas was calculated separately based on the estimated number of buildings in the area. An error margin of 5% was applied in calculating the sample sizes. In choosing the study areas for this research, a simple random sampling technique was employed. Through a balloting process, three regions—Ashanti, Greater Accra, and Eastern—were selected from the sixteen existing regions in Ghana. From each of these regions, one urban district was randomly chosen: Kumasi Metropolitan Assembly from the Ashanti region, Ga West Municipal from the Greater Accra region, and Akuapem North Municipal from the Eastern region. In each of the three study districts, a multi-stage stratified sampling procedure was used to select communities and property owners for the survey. In Ga West Municipality, seven (7) communities were randomly selected, and all of those communities were covered (see Table 1). In Akuapem North Municipality, the communities are grouped into six revenue collection zones by the assemblies, and these zones served as *strata* for sampling purposes. In the case of Kumasi Metropolis, the four (4) Sub-Metropolitan Districts served as strata. In Akuapem North and Kumasi, at least one community was selected from each stratum, depending on the number of communities in that stratum. The list of localities in each stratum then served as a sampling frame from which a number of localities were randomly selected. The respective sample sizes (number of sampled buildings) were then distributed proportionally among the selected communities based on their sizes (in terms of population and housing stock). Within the communities, buildings were randomly selected with the help of satellite images and ground-truthing [23]. The survey instruments were then administered to the owners (or their spouses) of those properties.

## Data collection

The data for the study was collected using a structured survey questionnaire. This was supplemented with field notebooks and cameras to aid direct field observations. The data collection

**Table 1. Distribution of sample size.**

| Akuapem North | | Ga West | | Kumasi | |
|---|---|---|---|---|---|
| Community | Sample size | Community | Sample size | Community | Sample size |
| Adawso | 30 | Amasaman | 43 | AbrepoKese | 39 |
| Tinkong | 22 | Fise | 90 | Ahodwo | 20 |
| Akropong | 78 | Hebron | 26 | Amakom | 47 |
| Larteh | 86 | Kuntunse | 62 | Asafo | 43 |
| Mamfe | 40 | Medie | 50 | Ash Town | 47 |
| Amanokrom | 30 | Pobiman/Pobi-Asaawa | 56 | Bantama | 47 |
| Mampong | 68 | Sarpeiman | 96 | Bohyen | 45 |
| Obosomase | 30 | - | - | Bompata | 26 |
| Osabene | 25 | - | - | Dichemso | 41 |
| - | - | - | - | Nhyiaeso | 18 |
| - | - | - | - | Santase | 27 |
| - | - | - | - | South Suntreso | 24 |
| **Total** | **409** | **Total** | **423** | **Total** | **424** |

exercise spanned three months, commencing on the 15th of January 2020 and ending on the 28th of April 2020. The survey questionnaire was structured under the following themes: background information and house owners' data, characteristics of property, sanitation facilities and practices available, and household income and expenditure. Primarily, the questionnaire was made up of closed-ended questions with a few open-ended questions designed to capture respondents' opinions and suggestions on some of the issues covered. The survey questionnaire was administered digitally using the Kobo Collect Application installed on enumerators' Android smart mobile phones. The Kobo Collect Application allowed the questionnaire written in a Microsoft Excel format to be uploaded onto Android mobile devices, allowing the enumerators to administer it to respondents and capture responses in a paperless manner. The collected data were then transmitted to an online database from where they were downloaded in Microsoft Excel formats for cleaning, validation, and analysis.

## Quality control

To ensure the accuracy and reliability of the data collected, several quality control measures were implemented throughout the study. Initially, the questionnaires were pre-tested and finalized before deployment in the field. This helped to address all grey areas not previously captured in the questionnaire to maximize outputs from the data collection exercise. The outcome of the pre-test was used to fine-tune the study tools and protocols where necessary. The study also put in place quality control measures to ensure that all the data collected from the research met acceptable standards. To do this, it was ensured that highly qualified teams of enumerators and supervisors were recruited for the study. The recruitment process took two weeks, starting from 1 December 2019 to 14 December 2019. After the recruitment process, two days of training was organized for the selected supervisors and enumerators. The training sessions involved presentations, discussions, field-based and hands-on demonstrations, and exercises using the tools and protocols developed for the study. Aside from the team leader, research officers with the requisite technical experience were assigned to field teams to ensure that the data collection process ran smoothly. Again, during the field data collection, each team, led by the research officers, met at the end of each day to summarize and evaluate data collection activities, identify gaps, and define ways to fill any gaps.

## Data analysis

The study used contingency tables, the chi-square test, and logistic regression methods to analyze the collected data. The study categorized the dependent variable, which represents ownership of a toilet facility, as (0 = no and 1 = yes). The independent variables consisted of the landlord's age, measured in years as a continuous variable; the landlord's sex, coded (0 = female and 1 = male); and the landlord's educational level, coded as (0 = no education, 1 = basic education, 2 = secondary education, and 3 = tertiary education). The landlord's occupation was categorized as follows: (0 = unemployed, 1 = trading/artisans/farming, and 2 = government/ private employee). The study quantified household size as a continuous variable, which represents the total number of individuals residing in the home. The coding for community categorization was coded as (0 represented low-class areas and 1 represented medium-class areas). The building's age was classified as (0 = 5–10 years, 1 = 11–20 years, 2 = 21–30 years, and 3 = 31 years and above). The study categorized the housing type as (0 = single-family house (improvised), 1 = multi-family house (semi-detached), and 2 = compound house (single story)). The construction materials used were categorized as (0 = for mud/wattle and daub work, 1 = for sandcrete (cement), and 2 = for burned bricks). The building's status (0 = not complete and 1 = completed). In monetary units, the study quantified household income and monthly spending as continuous variables. The insignificant regional differences observed in Models I, II, and III prompted the introduction of Model IV. Model IV considers the combined effect of the whole sample to assess the predictors of toilet facility ownership across the three areas because there were no regional variations regarding the dependent variable. This approach ensured that the study accurately represents the various socio-demographic and economic factors that influence toilet facility ownership. STATA 17 was used to analyze that data.

## Ethical considerations

In this research, key ethical considerations included informed voluntary participation, confidentiality, privacy, and data management. Prior to data collection, participants were thoroughly informed about the study and given the opportunity to ask questions regarding their involvement. Written and informed consent was obtained from all participants and informants as appropriate. Participation in the study was entirely voluntary, with no obligation for any individual to take part. To ensure anonymity and prevent any form of victimization, participants' responses were anonymized and assigned codes. Additionally, all data from interviews and questionnaires were securely stored and will be retained for five years before being discarded.

## Limitations of the study

While this study employed rigorous methodologies, several limitations related to the data collection process must be acknowledged. First, the reliance on self-reported data introduces the potential for response bias. As with any survey-based research, respondents may have been inclined to underreport or overreport certain variables such as household income, household expenditure, the state of sanitation facilities, or building characteristics, either due to social desirability bias or inaccuracies in recall. Self-reported income, for example, is often prone to misreporting, particularly in settings where respondents may have irregular or fluctuating incomes or where there may be sensitivity around disclosing financial information. This could affect the reliability of income as a predictor of toilet ownership. Second, the cross-sectional nature of the study limits the ability to establish causality. Although associations were identified between various factors (such as property characteristics, household income, and sanitation ownership), the study design does not allow for an examination of how these

relationships might evolve over time. Lastly, the study also faced challenges related to geographic representativeness. Although efforts were made to ensure a random and stratified sampling process, the study areas may not fully represent the diverse conditions in other urban districts across Ghana. Urban areas outside the selected municipalities could exhibit different socio-economic characteristics and sanitation practices, potentially limiting the generalizability of the findings.

## Results

### Descriptive statistics for the study

The analysis revealed that the gender of landlords does not exhibit a significant association with toilet ownership in Akuapem North ($\chi^2 = 0.341$, p = 0.559), Ga West ($\chi^2 = 0.011$, p = 0.917), and KMA ($\chi^2 = 3.30$, p = 0.069). The majority of landlords in all three regions were male: 61.1% in Ga West, 55.6% in KMA, and 56.6% in Akuapem North (see Table 2). Educational attainment of landlords showed significant correlations in KMA ($\chi^2 = 27.45$, p < 0.001) and Ga West ($\chi^2 = 28.35$, p < 0.001), but not in Akuapem North ($\chi^2 = 7.27$, p = 0.064). In Akuapem North, 30.1% of landlords had secondary education, whereas in KMA, 29.0% were without formal education. In Ga West, 36.3% had basic education.

Significant correlations were also observed between the occupation of landlords and toilet ownership in KMA ($\chi^2 = 19.97$, p < 0.001) and Ga West ($\chi^2 = 11.94$, p = 0.003), but not in Akuapem North ($\chi^2 = 1.85$, p = 0.396). In KMA, 211 (54.8%) landlords are engaged in trading, artisanal work, and farming. This was similar in Ga West (265; 62.8%) and Akuapem North (240; 57.3%). Community classification showed significant correlations with toilet ownership in all three regions: Akuapem North ($\chi^2 = 14.11$, p < 0.001), KMA ($\chi^2 = 22.52$, p < 0.001), and Ga West ($\chi^2 = 13.36$, p < 0.001). Notably, most landlords were located in medium-class areas: 78.3% in Akuapem North, 94.8% in KMA, and 70.4% in Ga West.

The age of buildings revealed significant relationships with toilet ownership in KMA ($\chi^2 = 10.12$, p = 0.018), Ga West ($\chi^2 = 38.91$, p < 0.001), with no significance in Akuapem North ($\chi^2 = 7.50$, p = 0.058). In Akuapem North, 53.2% of buildings were over 31 years old, whereas in KMA, 68.5% were over 31 years old, and in Ga West, 36.7% were 11–20 years old. The function of the building was significantly associated with toilet ownership in KMA ($\chi^2 = 10.66$, p = 0.005) but not in Akuapem North ($\chi^2 = 1.08$, p = 0.582) or Ga West ($\chi^2 = 3.59$, p = 0.17). In Akuapem North and Ga West, 90% and 83.9% of buildings were used for residential purposes only, respectively. In KMA, however, 71% of the buildings were for commercial use. There was a strong correlation between the type of housing and toilet ownership in all three regions: Akuapem North ($\chi^2 = 22.17$, p < 0.001), KMA ($\chi^2 = 22.41$, p < 0.001), and Ga West ($\chi^2 = 25.11$, p < 0.001). In Ga West, 48.8% were single-family houses, while in Akuapem North and KMA, 75.9% were compound houses. Building materials also exhibited significant correlations with toilet ownership in all regions: Akuapem North ($\chi^2 = 26.30$, p < 0.001), KMA ($\chi^2 = 42.93$, p < 0.001), and Ga West ($\chi^2 = 21.92$, p < 0.001). Sandcrete blocks were the predominant material: 91.9% in Akuapem North, 95.3% in KMA, and 96.4% in Ga West. Furthermore, the completion status of buildings showed significance in Akuapem North ($\chi^2 = 3.72$, p = 0.004) and Ga West ($\chi^2 = 22.79$, p < 0.001), but not in KMA ($\chi^2 = 0.46$, p = 0.496). The majority of buildings were completed: 88.8% in Akuapem North, 94% in KMA, and 78.4% in Ga West (Table 2).

Regarding household income, Akuapem North had the highest mean household income per month (GHC 1214.10, SD = 995.19), followed by Ga West (GHC 1605.47, SD = 1572.29), and KMA (GHC 667.12, SD = 744.91). Household expenditure was also highest in Akuapem North (GHC 1077.60, SD = 2108.51). The age of landlords showed less variation, with KMA

**Table 2. Cross-tabulation and chi-square analysis of respondent's demographic characteristics.**

| Socio-demographic characteristics | Akuapem north | X² (p-value) | KMA | X² (p-value) | Ga West | X² (p-alue) |
|---|---|---|---|---|---|---|
| | N = 410(%) | | N = 403(%) | | N = 432(%) | |
| **Ownership toilet facility** | | | | | | |
| No | 108(25.8) | | 108(26.8) | | 117(27.7) | |
| Yes | 302(72.1) | | 295(73.2) | | 305(72.3) | |
| **Sex of landlord** | | | | | | |
| Female | 173(41.3) | .341 (.559) | 179(44.4) | 3.30 (0.069) | 164(38.9) | .011 (.917) |
| Male | 237(56.6) | | 224(55.6) | | 258(61.1) | |
| **Educational level of landlord** | | | | | | |
| No education | 91(21.7) | 7.27 (.064) | 117(29.0) | 27.45 (< .001) | 65(15.4) | 28.35 (< .001) |
| Basic education | 124(29.6) | | 114(28.3) | | 153(36.3) | |
| Secondary education | 126(30.1) | | 96(23.8) | | 123(29.1) | |
| Tertiary education | 69(16.5) | | 76(18.9) | | 81(19.2) | |
| **Occupation of the Landlord** | | | | | | |
| Unemployed | 120(28.6) | 1.85 (.396) | 138(34.2) | 19.97 (< .001) | 58(13.7) | 11.94 (.003) |
| Trading/artisans/farming | 240(57.3) | | 211(54.8) | | 265(62.8) | |
| Government/private employee | 49(11.7) | | 18(4.5) | | 99(23.5) | |
| **Community classification** | | | | | | |
| Low class area | 82(19.6) | 14.11 (< .001) | 19(4.7) | 22.52 (< .001) | 125(29.6) | 13.36 (< .001) |
| Medium class area | 328(78.3) | | 382(94.8) | | 297(70.4) | |
| **Age of the building** | | | | | | |
| 5–10 years | 28(6.7) | 7.50 (.058) | 8(2.0) | 10.12 (.018) | 167(39.6) | 38.91 (< .001) |
| 11–20 years | 73(17.4) | | 46(11.4) | | 155(36.7) | |
| 21–30 years | 85(20.3) | | 73(18.1) | | 71(16.8) | |
| 31 years and above | 233(53.2) | | 276(68.5) | | 27(6.4) | |
| **Functions of the building** | | | | | | |
| Mixed use (residential and commercial) | 30(7.2) | 1.08 (.582) | 3(0.7) | 10.66 (.005) | 55(13) | 3.59 (0.17) |
| Commercial use only | 3(0.7) | | 286(71) | | 13(3.1) | |
| Residential use only | 377(90) | | 1(0.2) | | 354(83.9) | |
| **Type of house** | | | | | | |
| Singly family house (improvised) | 91(21.7) | | 62(15.4) | 22.41 (< .001) | 206(48.8) | |
| Multi-family house (semi-detached house) | 28(6.7) | 22.17 (< .001) | 35(8.7) | | 37(8.8) | 25.11 (< .001) |
| Compound house (single storey) | 257(61.3) | | 306(75.9) | | 111(26.3) | |
| **Material used to construct building** | | | | | | |
| Mud/wattle and daub work | 24(5.7) | 26.30 (< .001) | 14(3.5) | 42.93 (< .001) | 13(3.1) | 21.92 (< .001) |
| Sandcrete (cement blocks) | 385(91.9) | | 384(95.3) | | 407(96.4) | |
| Burns bricks | 1(0.2) | | 5(1.2) | | 1(0.2) | |
| **Nature of building completed** | | | | | | |
| Not completed | 38(9.1) | 3.72 (.054) | 24(6) | .46 (.496) | 90(21.3) | 22.79 (< .001) |
| Completed | 372(88.8) | | 379(94) | | 331(78.4) | |

having the highest mean age (60 years), followed by Akuapem North (56 years), and Ga West (48 years) (see Table 3).

## Inferential statistics

**Individual logistic regression.** In Table 4, the logistic regression study on toilet facility ownership in the Akuapem North, KMA, and Ga West areas identifies many significant

**Table 3. Summary statistics of household's income and expenditure.**

| Variable/ Parameter | Akuapem North | | KMA | | Ga-West | |
|---|---|---|---|---|---|---|
| | Mean | Standard deviation | Mean | Standard deviation | Mean | Standard deviation |
| Household income per month | 1214.10 | 995.19 | 667.12 | 744.91 | 1605.47 | 1572.29 |
| Household expenditure | 1077.60 | 869.67 | 305.88 | 314.62 | 1415.93 | 2108.51 |
| Age of landlord | 56.00 | 13.76 | 60 | 12.71 | 48 | 11.01 |

factors. For all three locations, the landlord's age has no significant impact on toilet facility ownership. The gender of the landlord does not also have a significant influence in Akuapem North, KMA, and Ga West on toilet facility ownership. The level of education has a significant effect on toilet facility ownership in these areas. Landlords with secondary education, are more likely to own a toilet facility in KMA (OR = 2.33, CI = [1.123–4.835]) than uneducated landlords. If a landlord has a tertiary education, they are more likely to own a toilet facility in both

**Table 4. Logistic regression on toilet facility ownership.**

| Ownership toilet facility | Akuapem North | | KMA | | Ga West | |
|---|---|---|---|---|---|---|
| | Model I | | Model II | | Model III | |
| | OR | CI | OR | CI | OR | CI |
| **Age of the landlord** | 1.005 | [.980–1.031] | 1.02 | [0.994–1.046] | 1.011 | [0.979–1.044] |
| **Sex of the landlord** | ref.(female) | | | | | |
| Male | 0.703 | [.378–1.309] | 1.118 | [0.687–0.649] | 0.76 | [0.389–1.485] |
| **Educational level** | ref. (no education) | | | | | |
| Basic education | 1.054 | [.471–2.360] | 1.349 | [.728–2.450] | 1.452 | [0.586–3.598] |
| Secondary education | 0.972 | [0.398–2.377] | 2.33** | [1.123–4.835] | 1.597 | [0.563–4.525] |
| Tertiary education | 3.47** | [.822–14.656] | 6.023*** | [1.832–19.794] | 2.294 | [0.526–10.005] |
| **Occupation of the landlord** | ref.(unemployed) | | | | | |
| Trading/artisans/farming | 1.794 | [.756–4.259] | 1.023 | [.528–1.981] | 0.766 | [0.254–2.309] |
| Government/private employee | 0.747 | [.191–2.922] | 3.279 | [.358–30.011] | 1.31 | [0.323–5.308] |
| **Household size** | 0.944 | [.846–1.053] | 1.07 | [.986–1.161] | 1.044 | [0.924–1.179] |
| **Community classification** | ref. (low class area) | | | | | |
| Medium class area | 2.018** | [1.030–3.953] | 3.486** | [1.009–12.052] | 1.846** | [0.917–3.715] |
| **Age of the building** | ref.(5-10years) | | | | | |
| 11–20 years | 5.042** | [1.044–24.350] | 4.852 | [.616–38.239] | 1.074 | [0.48–2.403] |
| 21–30 years | 2.241 | [.476–10.559] | 3.622 | [.517–25.381] | 0.469 | [0.179–1.230] |
| 31 years and above | 2.925 | [.617–13.869] | 2.17 | [.330–14.284] | .118*** | [0.032–0.438] |
| **Type of house** | ref. (Singly family house (improvised)) | | | | | |
| Multi-family (semi-detached house) | 1.94 | [.311–12.109] | 1.257 | [.1971–8.020] | 1.414 | [0.423–4.731] |
| Compound house (single storey) | .346** | [.135-.887] | .196*** | [.060-.645] | 0.621 | [0.288–1.339] |
| **The material used to build the house** | ref. (Mud/wattle and daub work) | | | | | |
| Sandcrete (cement) | 9.479*** | [3.085–29.131] | 3.31 | [.440–24.987] | 10.547*** | [1.815-61.289] |
| **Nature of the building** | ref. (not completed) | | | | | |
| Completed | 2.792** | [.992–7.856] | 0.705 | [.236–2.1092] | 3.783*** | [1.85–7.736] |
| **Household Income per month** | 1.001** | [1.0–1.001] | 1.00 | [.999–1.001] | 1.001*** | [1.00–1.001] |
| **Household expenditure per month** | 1.001 | [1.00–1.001] | 1.00 | [.998–1.002] | 1.00 | [1.00–1.00] |

*** p < .01, ** p < .05, * p < .1.

Akuapem North (OR = 3.47, CI = [0.822–14.656]) and in KMA (OR = 6.023, CI = [1.832–19.794]) than if they had no formal education.

The landlord's occupation, whether in commerce, artisan work, farming, or government or private employment, does not significantly influence the ownership of toilet facilities across different areas. Household size does not have a significant influence. Community categorization is shown to be a significant predictor, as landlords in medium-class regions are more likely to own toilet facilities in Akuapem North (OR = 2.018, CI = [1.030–3.953]), KMA (OR = 3.486, CI = [1.009–12.052]), and Ga West (OR = 1.846, CI = [0.917–3.715]) with higher odds as compared to landlords living in lower class areas. In certain areas, the age of the structure might serve as a significant indicator. In Akuapem North, buildings that are between 11 and 20 years old have a higher likelihood of having toilet facilities (OR = 5.042, CI = [1.044–24.350]) compared to those between 5–10 years old. On the other hand, in Ga West, structures that are 31 years old or older are less likely to have toilet facilities (OR = 0.118, CI = [0.032–0.438]), decreasing odds as compared to structures that are 5-10years old. The type of housing has a significant impact, since compound dwellings in Akuapem North (OR = 0.346, CI = [0.135–0.887]) and KMA (OR = 0.196, CI = [0.060–0.645]) have decreased odds of having toilet facilities in comparison to single family house.

The choice of construction material is a significant indicator since the use of sandcrete greatly increases the odds of having a toilet facility in Akuapem North (OR = 9.479, CI = [3.085–29.131]) and Ga West (OR = 10.547, CI = [1.815–61.289]) as compared to mud/wattle and daub work. Moreover, the completion status of buildings influences the presence of toilet facilities. Specifically, finished buildings in Akuapem North are 2.792 times more likely to have toilet facilities (CI = [0.992–7.856]) as compared to uncompleted houses, while completed buildings in Ga West are 3.783 times more likely to have toilet facilities ([1.85–7.736]). In Akuapem North, a higher household income has a marginal impact on the probability of having a toilet facility (OR = 1.001, CI = [1.0–1.001]). Similarly, in Ga West, a higher household income also has a marginal impact on the chance of owning a toilet facility (OR = 1.001, CI = [1.00–1.001]). However, household spending does not have a significant influence on toilet facility ownership in any area.

## Combined logistic regression

Table 5 shows the combined effect of the entire sample size on the dependent variable. Among the socio-demographic variables considered, the age of the landlord shows a statistically significant association with toilet facility ownership (OR = 1.014). Thus, for each one-year increase in the age of the landlord, there is a 1.4% increase in the likelihood of owning toilet facilities. Regarding the educational level of the landlord, those with secondary (OR = 1.752) and tertiary education (OR = 4.489) exhibit significantly higher odds of owning toilet facilities compared to those with no education. Similarly, community classification demonstrates a significant association, with landlords residing in medium-class areas exhibiting higher odds (OR = 2.013) of owning toilet facilities compared to those in low-class areas. The nature and characteristics of the housing structure also influence toilet facility ownership. Specifically, completed buildings are associated with significantly higher odds (OR = 2.625) of owning toilet facilities compared to incomplete structures. Moreover, buildings constructed with sandcrete (cement) have substantially higher odds (OR = 12.755) of having toilet facilities compared to those constructed with mud/wattle and daub work.

Additionally, household income exhibits a significant positive association (OR = 1.0) with toilet facility ownership, indicating that higher income levels are associated with a greater

**Table 5. Combined logistic regression model IV.**

| Ownership toilet facility | OR | St. Err. | p-value | [95% Conf Int.] |
|---|---|---|---|---|
| **Age of Landlord** | 1.014 | 0.008 | .053* | [1.00–1.029] |
| **Sex of landlord** | ref.(female) | | | |
| Male | 0.882 | 0.147 | 0.454 | [.636–1.224] |
| **Educational level of the landlord** | ref. (no education) | | | |
| Basic education | 1.305 | 0.267 | 0.193 | [.874–1.95] |
| Secondary education | 1.752 | 0.408 | .016** | [1.11–2.766] |
| Tertiary education | 4.489 | 1.621 | .00*** | [2.212–9.108] |
| **Occupation of the landlord** | ref.(unemployed) | | | |
| Trading/artisans/f~g | 1.002 | 0.225 | 0.992 | [.646–1.555] |
| Government employee | 0.999 | 0.396 | 0.998 | [.459–2.174] |
| Private employee | 3.751 | 3.982 | 0.213 | [.468–30.045] |
| Household size | 1.028 | 0.027 | 0.302 | [.976–1.082] |
| **Community Classification** | ref. (low class area) | | | |
| Medium class area | 2.013 | 0.417 | .001*** | [1.342–3.021] |
| **Age of the building** | ref.(5-10years) | | | |
| 11–20 years | 1.567 | 0.499 | 0.158 | [.839–2.926] |
| 21–30 years | 1.002 | 0.328 | 0.996 | [.527–1.905] |
| 31 years and above | 0.844 | 0.274 | 0.602 | [.447–1.594] |
| **Type of house** | ref. (Singly family house (improvised) | | | |
| Multi-family house (semi-detached house) | 1.391 | 0.566 | 0.417 | [.627–3.086] |
| Compound house (single storey) | 0.401 | 0.096 | .00*** | [.251-.641] |
| **Material used to construct the house** | ref. (Mud/wattle and daub work) | | | |
| Sandcrete (cement) | 12.755 | 5.369 | 0.00*** | [5.59–29.106] |
| Burns bricks | 4.223 | 4.061 | 0.134 | [.641–27.812] |
| **Nature of building** | ref. (not completed) | | | |
| Completed | 2.625 | 0.631 | .00*** | [1.64–4.204] |
| Household income per month | 1.00 | 0.00 | .026** | [1.00–1.001] |
| Household expenditure per month | 1.00 | 0.00 | .50* | [1.00–1.00] |

*** p < .01, ** p < .05, * p < .1.

likelihood of having access to sanitation facilities. However, household expenditure does not demonstrate a significant association with toilet facility ownership (OR = 1.0).

## Discussions

According to the study, toilet ownership among property owners in the urban cities of Ghana is influenced by factors such as the age of the house owner, their education level, community classification, type of house, materials used to construct the house, nature of the building, and household monthly income.

The study finds a significant association between age and toilet ownership, with older property owners more likely to invest in sanitation facilities. This finding aligns with earlier studies conducted in Indonesia, India, and certain parts of Ghana (Dodowa and Cape Coast), where older household heads were more likely to own and adopt household toilets [24–27]. This can be understood through the lens of health awareness and financial capacity. Older property owner may have a heightened awareness of the health risks posed by poor sanitation, which drives their decision to invest in toilet facilities. This awareness is particularly relevant in

regions where open defecation remains prevalent [19,28]. The findings align with existing literature that emphasizes the vulnerability of older adults to sanitation-related health risks and their subsequent motivation to invest in improved sanitation [19,29–32]. Financially, older property owners might have accumulated more wealth over time, providing them with the necessary resources to invest in sanitation infrastructure. While financial stability is a core driver, the decision to invest in toilets can also reflect an individual's life stage, where retirement or health conditions may prompt a greater focus on personal sanitation [29–31]. However, Onyeabor and Umeh [33] found contrasting results. In their study conducted in Ebonyi State in Nigeria, an inverse relationship between age and access to improved toilets was observed. This notwithstanding, the overall trend suggests that older property owners are generally more likely to prioritize and invest in improved sanitation facilities, driven by heightened health awareness and financial stability.

Education is one of the most consistent predictors of improved sanitation outcomes globally.

Educated property owners are more likely to have access to information about the health and economic benefits of proper sanitation. They also tend to better understand the long-term cost savings of investing in sanitation, as poor hygiene and open defecation can result in significant health-related costs [34]. Our findings show that property owners with secondary and tertiary education have higher odds of toilet ownership, aligning with existing studies that highlight the role of education in promoting sanitation adoption and sustainability [17,19,25,27,34–37]. A study in the Bole District of Ghana revealed that individuals with tertiary education were approximately six times more likely to own a toilet than those without formal education [38]. Similarly, studies by Armah et al. [39] and Nyambe et al. [40] found that higher educational attainment was linked to increased access to improved sanitation. Rodger et al. [41] also identified a strong correlation between education level and latrine ownership, further supporting our observation that more educated landlords are more inclined to provide toilet facilities on their properties. Education also contributes to the broader understanding of sanitation beyond mere access to toilets. Munamati et al. [37] demonstrated that countries with higher education levels had made significant progress in improving sanitation access, as education builds the necessary capacities for the effective implementation and maintenance of sanitation systems. Investing in education, therefore, holds significant potential for advancing the achievement of Sustainable Development Goal 6 (SDG 6) by empowering individuals to adopt and sustain sanitation solutions.

Community classification emerged as a significant predictor of toilet ownership, with landlords in medium-class areas having higher odds (OR = 2.013) of providing toilet facilities compared to those in low-class areas. This disparity is largely driven by economic capacity and rental incentives. Landlords in medium-class areas typically have greater financial resources, enabling them to invest in the construction and maintenance of sanitation infrastructure. In contrast, landlords in low-class areas often face financial constraints that limit their ability to provide such facilities. This economic inequality mirrors findings from Prüss-Ustün et al [30], who noted that residents in low-income settings often struggle with inadequate water, sanitation, and hygiene (WASH) infrastructure due to limited financial resources. Mara et al [2] also emphasize that affordability is a major barrier to improved sanitation in low-income communities. While aspirations for better sanitation systems exist among residents of low-income settlements [42], the high initial investment costs present a significant challenge for property owners [43,44]. Additionally, there is a clear economic incentive for landlords in medium-class areas to provide toilets, as doing so increases the attractiveness of their properties to potential tenants. Providing toilet facilities may lead to higher rents or attract a wider pool of potential renters. This incentive, however, is less pronounced in low-income areas, where

space and financial limitations make it difficult for property owners to afford household sanitation. Further, in low-income neighborhoods, access to improved sanitation is often deprioritized by renters. Individuals in these areas primarily focus on securing accommodation [42], and the availability of toilet facilities becomes secondary [13]. Consequently, landlords in these areas may perceive little benefit in investing in toilet facilities, as it may not provide a competitive advantage in the rental market.

Again, the findings reveal a significant association between the nature and characteristics of a building or house and the likelihood of owning toilet facilities. Our findings show that completed buildings are significantly more likely (OR = 2.625) to have toilet facilities compared to incomplete structures. Moreover, buildings constructed with sandcrete (cement) exhibit markedly higher odds (OR = 12.755) of having toilet facilities than those built with mud or wattle and daub. These findings carry broader socioeconomic implications. The strong association between completed sandcrete buildings and toilet ownership suggests a potential overlap with socioeconomic status. Wealthier individuals are more likely to afford completed, permanent dwellings constructed with sandcrete, which, in turn, are more likely to have toilets. Sandcrete structures are generally associated with higher construction costs and are considered more durable and permanent compared to mud dwellings. Therefore, landlords or property owners may be more inclined to invest in toilet facilities in such structures perceived as long-term investments. In contrast, incomplete structures, often lacking essential features like roofs, windows, walls, plastering, or painting, may represent unfinished or temporary shelters. Thus, these structures are typically prioritized for temporary or transitional habitation rather than permanent dwellings. Consequently, the likelihood of investing in sanitation facilities in such buildings is lower. These findings also highlight a critical yet often overlooked aspect of sanitation access in Ghana. They reveal issues of weak enforcement of sanitation by-laws by local assemblies. In Ghana, the local regulations (by-laws) of various districts require property owners to obtain a design and construction permit before building houses, and a habitation (occupancy) permit upon completion. The local authorities are responsible for ensuring that adequate sanitation facilities are incorporated into the building designs and are present in completed buildings before issuing permits. Despite the requirement for improved sanitation as a condition for building permits in Ghana, these standards are not consistently enforced [42,45,46]. The lack of strict enforcement of these by-laws contributes to the prevalence of incomplete structures and buildings without adequate sanitation facilities. Strengthening the enforcement of sanitation by-laws and ensuring compliance with building regulations could significantly improve access to sanitation facilities.

Another clear finding that emerged from the study is the relationship between household income and toilet ownership. There is a positive correlation between households' monthly income and the likelihood of owning toilet facilities. This finding is consistent with a substantial body of literature that underscores the critical role of socioeconomic status in determining access to sanitation facilities. Increased household income often correlates with improved living standards, which include better housing and access to utilities. Higher household income translates into greater financial capacity to construct and maintain toilet facilities. For instance, studies by Mara et al. [2] and Mara and Evans [47] found that income is a key determinant of a household's ability to afford the initial costs of toilet construction and the ongoing expenses of maintenance and repair. This is further supported by O'Connell [34], who found that households with improved latrines are generally wealthier than those who own unimproved latrines or practice open defecation. This trend is consistent across multiple studies, with lower-income households consistently facing significant barriers to toilet ownership due to financial limitations [25]. Jenkins and Scott [48] and Hirai et al [49] also found that lower-income households are less likely to own toilets, as they struggle with the upfront costs of construction

and the recurring expenses for maintenance. These financial constraints make it difficult for households in the lower socioeconomic quintiles to invest in sanitation facilities, contributing to persistent disparities in access to improved sanitation.

## Conclusion

Improving toilet ownership is crucial for reducing sanitation-related inequalities and advancing progress towards universal access to adequate and equitable sanitation, particularly for vulnerable populations. This study demonstrates that key factors such as the age and education level of property owners, community classification, building characteristics, and household income significantly influence toilet ownership in urban areas of Ghana. Older and more educated property owners, as well as those with higher incomes, are more likely to own toilets. Likewise, toilets are more common in completed buildings, properties made of durable materials like sandcrete, and in medium-class communities. These disparities highlight the broader socio-economic inequalities in access to sanitation facilities and underscore the need for improved enforcement of sanitation regulations.

To address these challenges, we recommend stricter enforcement of building codes and regulations that mandate the provision of toilet facilities in all residential and commercial properties. Metropolitan, municipal, and district assemblies must be adequately resourced to enhance enforcement mechanisms through staff training, regular monitoring, the use of technology, and community engagement. Public awareness campaigns and community-level initiatives can further create social pressure to drive compliance. Additionally, a clear legal framework that protects tenant rights to basic sanitation and enforces swift penalties for violations would support the effective implementation of sanitation regulations.

Addressing financial barriers to toilet ownership is equally critical. Policymakers should explore innovative sanitation financing solutions in urban areas, such as microfinance schemes that offer low-interest loans or flexible payment plans for property owners to construct toilet facilities. This could be facilitated through partnerships between local governments, financial institutions, and NGOs to enable property owners, especially in low-income areas, to afford the upfront costs of construction. Another approach could be revolving funds, where beneficiaries repay loans over time, allowing funds to be reinvested in new sanitation projects.

## Supporting information

**S1 Table. Types of toilet facilities owned by property owners.**
(DOCX)

**S2 Table. Where households without toilet facilities defecate.**
(DOCX)

**S3 Table. Respondent's level of satisfaction with public toilets.**
(DOCX)

## Acknowledgments

We are grateful to Sharon Bonso for her assistance at various stages of the manuscript preparation. We also appreciate the officials of the various assemblies for their assistance and the cooperative respondents who participated in the research. Finally, we are very grateful to the enumerators and field supervisors for their participation in the data collection.

## Author Contributions

**Conceptualization:** Godwin Armstrong Duku, Nana Kobea Bonso, Eugene Appiah-Effah, Kwabena Biritwum Nyarko.

**Data curation:** Godwin Armstrong Duku, Nana Kobea Bonso.

**Formal analysis:** Godwin Armstrong Duku, Nana Kobea Bonso, Yenube Clement Kunkuaboor, Emmanuel Nouwati.

**Funding acquisition:** Eugene Appiah-Effah, Kwabena Biritwum Nyarko.

**Investigation:** Godwin Armstrong Duku, Nana Kobea Bonso, Eugene Appiah-Effah, Yenube Clement Kunkuaboor, Emmanuel Nouwati, Barbara Gyapong-Korsah, Ebenezer Acquah, Kwabena Biritwum Nyarko.

**Methodology:** Godwin Armstrong Duku, Yenube Clement Kunkuaboor, Emmanuel Nouwati.

**Project administration:** Eugene Appiah-Effah, Barbara Gyapong-Korsah, Ebenezer Acquah, Kwabena Biritwum Nyarko.

**Resources:** Eugene Appiah-Effah, Barbara Gyapong-Korsah, Ebenezer Acquah, Kwabena Biritwum Nyarko.

**Software:** Godwin Armstrong Duku, Yenube Clement Kunkuaboor, Emmanuel Nouwati.

**Supervision:** Eugene Appiah-Effah, Kwabena Biritwum Nyarko.

**Validation:** Godwin Armstrong Duku, Nana Kobea Bonso.

**Visualization:** Godwin Armstrong Duku, Nana Kobea Bonso.

**Writing – original draft:** Godwin Armstrong Duku, Nana Kobea Bonso.

**Writing – review & editing:** Godwin Armstrong Duku, Nana Kobea Bonso, Eugene Appiah-Effah, Yenube Clement Kunkuaboor, Emmanuel Nouwati, Barbara Gyapong-Korsah, Ebenezer Acquah, Kwabena Biritwum Nyarko.

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
