## [Decision Letter · Decision Letter 0]

10 Sep 2024

PONE-D-24-27907Achieving Universal Sanitation in Ghana: An Analysis of Key Drivers of Toilet Ownership Among Property Owners in Urban Areas.PLOS ONE

Dear Dr. Duku,

Thank you for submitting your manuscript to PLOS ONE. After careful consideration, we feel that it has merit but may require minor revisions to meet PLOS ONE’s publication criteria as it currently stands. Therefore, we invite you to submit a revised version of the manuscript that addresses the points raised during the review process.

Please submit your revised manuscript by Oct 25 2024 11:59PM. If you will need more time than this to complete your revisions, please reply to this message or contact the journal office at plosone@plos.org. Please include the following items when submitting your revised manuscript:A rebuttal letter that responds to each point raised by the academic editor and reviewer(s). You should upload this letter as a separate file labeled 'Response to Reviewers'.A marked-up copy of your manuscript that highlights changes made to the original version. You should upload this as a separate file labeled 'Revised Manuscript with Track Changes'.An unmarked version of your revised paper without tracked changes. You should upload this as a separate file labeled 'Manuscript'.If applicable, we recommend that you deposit your laboratory protocols in protocols.io to enhance the reproducibility of your results. Protocols.io assigns your protocol its own identifier (DOI) so that it can be cited independently in the future. For instructions see: https://journals.plos.org/plosone/s/submission-guidelines#loc-laboratory-protocols. Additionally, PLOS ONE offers an option for publishing peer-reviewed Lab Protocol articles, which describe protocols hosted on protocols.io. Read more information on sharing protocols at https://plos.org/protocols?utm_medium=editorial-email&utm_source=authorletters&utm_campaign=protocols.

We look forward to receiving your revised manuscript.

Kind regards,

Elmond Bandauko, PhD 

Academic Editor

PLOS ONE

Journal Requirements: When submitting your revision, we need you to address these additional requirements. 1. Please ensure that your manuscript meets PLOS ONE's style requirements, including those for file naming. The PLOS ONE style templates can be found at https://journals.plos.org/plosone/s/file?id=wjVg/PLOSOne_formatting_sample_main_body.pdf and https://journals.plos.org/plosone/s/file?id=ba62/PLOSOne_formatting_sample_title_authors_affiliations.pdf 2. We note that your Data Availability Statement is currently as follows: All relevant data are within the manuscript and its Supporting Information files. Please confirm at this time whether or not your submission contains all raw data required to replicate the results of your study. Authors must share the “minimal data set” for their submission. PLOS defines the minimal data set to consist of the data required to replicate all study findings reported in the article, as well as related metadata and methods (https://journals.plos.org/plosone/s/data-availability#loc-minimal-data-set-definition). For example, authors should submit the following data: - The values behind the means, standard deviations and other measures reported;- The values used to build graphs;- The points extracted from images for analysis. Authors do not need to submit their entire data set if only a portion of the data was used in the reported study. If your submission does not contain these data, please either upload them as Supporting Information files or deposit them to a stable, public repository and provide us with the relevant URLs, DOIs, or accession numbers. For a list of recommended repositories, please see https://journals.plos.org/plosone/s/recommended-repositories. If there are ethical or legal restrictions on sharing a de-identified data set, please explain them in detail (e.g., data contain potentially sensitive information, data are owned by a third-party organization, etc.) and who has imposed them (e.g., an ethics committee). Please also provide contact information for a data access committee, ethics committee, or other institutional body to which data requests may be sent. If data are owned by a third party, please indicate how others may request data access. 3. Please review your reference list to ensure that it is complete and correct. If you have cited papers that have been retracted, please include the rationale for doing so in the manuscript text, or remove these references and replace them with relevant current references. Any changes to the reference list should be mentioned in the rebuttal letter that accompanies your revised manuscript. If you need to cite a retracted article, indicate the article’s retracted status in the References list and also include a citation and full reference for the retraction notice. 

Additional Editor Comments:

Dear Dr. Godwin Armstrong Duku

The reviewers have submitted their reports. However, one of the reviewers said they could not review the paper because there was no introduction. Did you encounter any challenges when submitting your paper? I am therefore, giving you an opportunity to resubmit your paper following minor revisions. You will see the comments from reviewer 1.

Reviewers' comments:

Reviewer's Responses to Questions

**Comments to the Author**

1. Is the manuscript technically sound, and do the data support the conclusions?

Reviewer #1: Yes

Reviewer #2: Partly

2. Has the statistical analysis been performed appropriately and rigorously? 

Reviewer #1: Yes

Reviewer #2: N/A

3. Have the authors made all data underlying the findings in their manuscript fully available?

Reviewer #1: Yes

Reviewer #2: No

4. Is the manuscript presented in an intelligible fashion and written in standard English?

Reviewer #1: Yes

Reviewer #2: Yes

5. Review Comments to the Author

Reviewer #1: Achieving Universal Sanitation in Ghana: An Analysis of Key Drivers of Toilet Ownership Among Property Owners in Urban Areas.

Comments

The paper addresses an important public health issue in urban Ghana, focusing on the determinants of household toilet ownership.

The manuscript is generally well-structured, with a clear flow from the introduction through to the conclusion. However, in my review I did not see the introduction section. A technical error, I guess. The introduction could benefit from a more comprehensive literature review that situates the study within the context of existing research on urban sanitation in Ghana and other similar contexts – If the authors have not adequately done so.

The authors should provide more justification for the choice of the three urban districts selected for the study. Were these districts chosen because they are representative of urban areas in Ghana, or were there other reasons?

It would also be beneficial to discuss any potential limitations related to the data collection process, such as response bias or the accuracy of self-reported data on household income and building characteristics.

The discussion appropriately connects the results to the broader context of urban sanitation in Ghana. However, it would benefit from a more critical analysis of the findings. For instance, the paper suggests that "enforcing building regulations requiring toilet facilities in all properties is crucial." Could the authors provide more concrete examples of how such regulations could be enforced in practice?

The suggestion for "innovative sanitation financing interventions" is a good one, but the paper could delve deeper into what these interventions might look like, particularly in the context of urban Ghana where financial constraints are a significant barrier.

The conclusion succinctly summarizes the key findings and recommendations. However, the authors could enhance the impact of their conclusions by explicitly linking them to the Sustainable Development Goals (SDGs), particularly SDG 6 (Clean Water and Sanitation).

Authors indicated that “The survey was supplemented with field notebooks and cameras to aid direct field observations.” I did not see results reflective of that, or at least images/ pictures of observations from the field.

In the discussion from line 6756 provide more specific details of the studies you are engaging with……eg. For example, a study in Ghana found that…. where exactly in Ghana?…

Overall Recommendation: The paper addresses a critical issue in urban sanitation and provides valuable insights into the determinants of toilet ownership among property owners in Ghana. With some revisions of suggested areas for improvement, the manuscript could make a significant contribution to the field and be a valuable resource for policymakers. I recommend minor revisions before the manuscript is considered for publication.

Reviewer #2: I HAVE NOT UNDERTAKEN A REVIEW OF THIS MANUSCRIPT BECAUSE THERE IS NO INTRODUCTION SECTION IN THE ATTACHMENTS. CAN I KINDLY HAVE THE COMPLETE MANUSCRIPT, POSSIBLY IN WORD FORMAT.

WOULD APPRECIATE THE CORRECT DOCUMENT BEING SHARED

6. PLOS authors have the option to publish the peer review history of their article (what does this mean?). If published, this will include your full peer review and any attached files.

Reviewer #1: No

Reviewer #2: No

---

## [Author Response · Author response to Decision Letter 0]

11 Oct 2024

Achieving Universal Sanitation in Ghana: An Analysis of Key Drivers of Toilet Ownership Among Property Owners in Urban Areas.

Comments

The paper addresses an important public health issue in urban Ghana, focusing on the determinants of household toilet ownership.

1. The manuscript is generally well-structured, with a clear flow from the introduction through to the conclusion. However, in my review, I did not see the introduction section. A technical error, I guess. The introduction could benefit from a more comprehensive literature review that situates the study within the context of existing research on urban sanitation in Ghana and other similar contexts – If the authors have not adequately done so.

Response: We sincerely appreciate your thoughtful feedback and apologize for the technical error. The introduction was indeed included in the original submission, and we have ensured that it is now properly visible in the revised manuscript. We believe we have undertaken a comprehensive literature review to situate the study within the broader context of urban sanitation in Ghana and similar contexts. We have thoroughly gone through the introduction to improve the flow and coherence of our argument as per your suggestion.

2. The authors should provide more justification for the choice of the three urban districts selected for the study. Were these districts chosen because they are representative of urban areas in Ghana, or were there other reasons?

Response: Thank you for highlighting this. We selected the three districts (Kumasi Metropolitan Assembly, Ga West Municipal, and Akuapem North Municipal) using a random sampling method, and we have now included a detailed explanation of this process in the methodology section. We have also provided justification in the introduction section that these districts, though randomly chosen, represent different levels of urbanization in Ghana, thus providing a diverse urban context for the study.

3. It would also be beneficial to discuss any potential limitations related to the data collection process, such as response bias or the accuracy of self-reported data on household income and building characteristics.

Response: We acknowledge the importance of discussing limitations and have now expanded the section on data limitations. Specifically, we have addressed potential response biases and the accuracy of self-reported data.

4. The discussion appropriately connects the results to the broader context of urban sanitation in Ghana. However, it would benefit from a more critical analysis of the findings. For instance, the paper suggests that "enforcing building regulations requiring toilet facilities in all properties is crucial." Could the authors provide more concrete examples of how such regulations could be enforced in practice?

Response: We appreciate this insightful comment. We have expanded our discussion to provide more concrete examples of how building regulations requiring toilet facilities in all properties could be enforced in practice. For instance, we now discuss specific mechanisms such as routine building inspections, stricter penalties for non-compliance, and the potential for leveraging local government partnerships to ensure adherence.

5. The suggestion for "innovative sanitation financing interventions" is a good one, but the paper could delve deeper into what these interventions might look like, particularly in the context of urban Ghana where financial constraints are a significant barrier.

Response: Thank you for this suggestion. We have expanded our discussion on potential innovative sanitation financing interventions, particularly within the context of urban Ghana. We now provide examples such as microfinance schemes for toilet construction and revolving fund to overcome financial barriers.

6. The conclusion succinctly summarizes the key findings and recommendations. However, the authors could enhance the impact of their conclusions by explicitly linking them to the Sustainable Development Goals (SDGs), particularly SDG 6 (Clean Water and Sanitation).

Response: We agree with this recommendation and have revised the conclusion to explicitly link our findings to SDG 6 (Clean Water and Sanitation).

7. Authors indicated that “The survey was supplemented with field notebooks and cameras to aid direct field observations.” I did not see results reflective of that, or at least images/ pictures of observations from the field.

Response: We apologize for the oversight. We have now incorporated images from our field observations into the method section.

8. In the discussion from line 6756 provide more specific details of the studies you are engaging with……eg. For example, a study in Ghana found that…. where exactly in Ghana?…

Response: We have revised the discussion to include more specific details regarding the studies referenced. For example, we now explicitly mention the regions or cities in Ghana where particular studies were conducted to allow for greater clarity and contextual relevance. 

9. Overall Recommendation: The paper addresses a critical issue in urban sanitation and provides valuable insights into the determinants of toilet ownership among property owners in Ghana. With some revisions of suggested areas for improvement, the manuscript could make a significant contribution to the field and be a valuable resource for policymakers. I recommend minor revisions before the manuscript is considered for publication.

Response: We sincerely appreciate the positive evaluation of our manuscript. We have addressed all the points raised to the best of our ability and believe that the revisions strengthen the overall quality and clarity of the paper.

---

## [Editor Report · Decision Letter 1]

26 Nov 2024

Achieving Universal Sanitation in Ghana: An Analysis of Key Drivers of Toilet Ownership Among Property Owners in Urban Areas.

PONE-D-24-27907R1

Dear Dr. Armstrong

We’re pleased to inform you that your manuscript has been judged scientifically suitable for publication and will be formally accepted for publication once it meets all outstanding technical requirements.

Kind regards,

Elmond Bandauko, PhD

Academic Editor

PLOS ONE

Additional Editor Comments (optional):

Dear Dr. Armstrong

Thank you for submitting your revised manuscript "Achieving Universal Sanitation in Ghana: An Analysis of Key Drivers of Toilet Ownership Among Property Owners in Urban Areas". I am pleased to inform you that your paper has been accepted for publication.

Looking forward to your future submissions
---

## [Editor Report · Acceptance letter]

10 Dec 2024

PONE-D-24-27907R1 

PLOS ONE

Dear Dr. Duku, 

I'm pleased to inform you that your manuscript has been deemed suitable for publication in PLOS ONE. Congratulations! Your manuscript is now being handed over to our production team.

Kind regards, 

on behalf of

Dr Elmond Bandauko 

Academic Editor

PLOS ONE